# A Paradigmatic Approach to Find the Valency-Based *K*-Banhatti and Redefined Zagreb Entropy for Niobium Oxide and a Metal–Organic Framework

**DOI:** 10.3390/molecules27206975

**Published:** 2022-10-17

**Authors:** Muhammad Usman Ghani, Faisal Sultan, El Sayed M. Tag El Din, Abdul Rauf Khan, Jia-Bao Liu, Murat Cancan

**Affiliations:** 1Institute of Mathematics, Khawaja Fareed University of Engineering & Information Technology, Abu Dhabi Road, Rahim Yar Khan 64200, Pakistan; 2Center of Research, Faculty of Engineering, Future University in Egypt, New Caira 11835, Egypt; 3Department of Mathematics, Faculty of Science, Ghazi University, Dera Ghazi Khan 32200, Pakistan; 4School of Mathematics and Physics, Anhui Jianzhu University, Hefei 230601, China; 5Faculty of Education, Yuzuncu Yil University, Van 65140, Turkey

**Keywords:** molecular graph, niobium oxide, metal–organic framework, topological indices, K-Banhatti entropies, redefined Zagreb entropies, Atom–bond sum connectivity entropy

## Abstract

Entropy is a thermodynamic function in chemistry that reflects the randomness and disorder of molecules in a particular system or process based on the number of alternative configurations accessible to them. Distance-based entropy is used to solve a variety of difficulties in biology, chemical graph theory, organic and inorganic chemistry, and other fields. In this article, the characterization of the crystal structure of niobium oxide and a metal–organic framework is investigated. We also use the information function to compute entropies by building these structures with degree-based indices including the *K*-Banhatti indices, the first redefined Zagreb index, the second redefined Zagreb index, the third redefined Zagreb index, and the atom-bond sum connectivity index.

## 1. Introduction

The optical properties of metallic nanoparticles have drawn the attention of scientists and researchers. The heat created by the nanoparticles overwhelms cancer tissue while causing no harm to healthy cells. Niobium nanoparticles have the capacity to easily attach to ligands, making them ideal for optothermal cancer treatment. Chemical graph theory is a contemporary branch of applied chemistry, which has remained an attractive area of research for scientists during the past two decades, and significant contributions have been made by scientists in this area of research including [1,2,3,4,5,6,7]. We investigate the relationship between atoms and bonds using combinatorial approaches such as vertex and edge partitions. Topological indices are essential in providing directions for treating malignancies or tumors. These indices can be obtained experimentally or numerically. Although experimental data are valuable, they are also costly; therefore, computational analysis gives a cost-effective and time-efficient solution.

The transformation of a chemical structure into a number is used to generate a topological index. The topological index is a graph invariant that characterizes the graph’s topology while remaining invariant throughout graph automorphism. A topological index is a numerical number defined only by the graph. The eccentricity-based topological indices are crucial in chemical graph theory [8]. Wiener, a chemist, first used a topological index in 1947 while researching the relationship between the molecular structure and the physical and chemical properties of certain hydrocarbon compounds [9]. In 2010, Damir et al. defined the redefined second Zagreb index as the same as the inverse sum indeg index [10].

We used the concept of valency-based entropies in this article, where $v_{{\dot{a}}_{1}}$ and $v_{{\dot{a}}_{2}}$ denote the valency of atoms, ${\dot{a}}_{1}$ and ${\dot{a}}_{2}$, within the molecule. Kulli started computing valency-based topological indices in 2016 using the valency of atom bonds and some Banhatti indices [11,12,13], each of which has the following definition:

The first valence-based *K*-Banhatti polynomial and the first *K*-Banhatti index are as follows:(1)$$\;B_{1}\left(G,x\right)=\sum_{{\dot{a}}_{1}∼{\dot{a}}_{2}}x^{\left(v_{{\dot{a}}_{1}}+v_{{\dot{a}}_{2}}\right)}\;\;\;\;\;\;B_{1}\left(G\right)=\sum_{{\dot{a}}_{1}∼{\dot{a}}_{2}}\left(v_{{\dot{a}}_{1}}+v_{{\dot{a}}_{2}}\right)$$

The second valence based *K*-Banhatti polynomial and the second *K*-Banhatti index are as follows, respectively:(2)$$\;B_{2}\left(G,x\right)=\sum_{{\dot{a}}_{1}∼{\dot{a}}_{2}}x^{\left(v_{{\dot{a}}_{1}}\times v_{{\dot{a}}_{2}}\right)}\;\;\;\;\;\;B_{2}\left(G\right)=\sum_{{\dot{a}}_{1}∼{\dot{a}}_{2}}\left(v_{{\dot{a}}_{1}}\times v_{{\dot{a}}_{2}}\right)$$

The first valence based hyper *K*-Banhatti polynomial and the firstst hyper *K*-Banhatti index are as follows, respectively:(3)$$\;HB_{1}\left(G,x\right)=\sum_{{\dot{a}}_{1}∼{\dot{a}}_{2}}x^{{\left(v_{{\dot{a}}_{1}}+v_{{\dot{a}}_{2}}\right)}^{2}}\;\;\;\;\;\;HB_{1}\left(G\right)=\sum_{{\dot{a}}_{1}∼{\dot{a}}_{2}}{\left(v_{{\dot{a}}_{1}}+v_{{\dot{a}}_{2}}\right)}^{2}$$
(4)$$\;HB_{2}\left(G,x\right)=\sum_{{\dot{a}}_{1}∼{\dot{a}}_{2}}x^{{\left(v_{{\dot{a}}_{1}}\times v_{{\dot{a}}_{2}}\right)}^{2}}\;\;\;\;\;\;HB_{2}\left(G\right)=\sum_{{\dot{a}}_{1}∼{\dot{a}}_{2}}{\left(v_{{\dot{a}}_{1}}\times v_{{\dot{a}}_{2}}\right)}^{2}$$

In 2013, Ranjini [14] introduced a redefined version of the Zagreb indices $ReZG_{1}$, and in 2021, Shanmukha [15] defined them as
(5)$$\;\;ReZG_{1}\left(G,x\right)=\sum_{{\dot{a}}_{1}∼{\dot{a}}_{2}}x^{\frac{v_{{\dot{a}}_{1}}+v_{{\dot{a}}_{2}}}{v_{{\dot{a}}_{1}}\times v_{{\dot{a}}_{2}}}}\;\;\;\;\;\;ReZG_{1}=\sum_{{\dot{a}}_{1}∼{\dot{a}}_{2}}\frac{v_{{\dot{a}}_{1}}+v_{{\dot{a}}_{2}}}{v_{{\dot{a}}_{1}}\times v_{{\dot{a}}_{2}}}.$$
(6)$$\;\;ReZG_{2}\left(G,x\right)=\sum_{{\dot{a}}_{1}∼{\dot{a}}_{2}}x^{\frac{v_{{\dot{a}}_{1}}\times v_{{\dot{a}}_{2}}}{v_{{\dot{a}}_{1}}+v_{{\dot{a}}_{2}}}}\;\;\;\;\;\;ReZG_{2}=\sum_{{\dot{a}}_{1}∼{\dot{a}}_{2}}\frac{v_{{\dot{a}}_{1}}\times v_{{\dot{a}}_{2}}}{v_{{\dot{a}}_{1}}+v_{{\dot{a}}_{2}}}.$$

The third redefined Zagreb index was defined as
(7)$$\;ReZG_{3}\left(G,x\right)=\sum_{{\dot{a}}_{1}∼{\dot{a}}_{2}}x^{\left(v_{{\dot{a}}_{1}}\times v_{{\dot{a}}_{2}}\right)\left(v_{{\dot{a}}_{1}}+v_{{\dot{a}}_{2}}\right)}\;\;ReZG_{3}=\sum_{{\dot{a}}_{1}∼{\dot{a}}_{2}}\left(v_{{\dot{a}}_{1}}\times v_{{\dot{a}}_{2}}\right)\left(v_{{\dot{a}}_{1}}+v_{{\dot{a}}_{2}}\right)$$

Recently, Ali et al. amalgamated the atom-bond connectivity index and sum connectivity index and initiated the new molecular descriptor named the atom-bond sum-connectivity index [16], defined as:(8)$$\;ABS\left(G,x\right)=\sum_{{\dot{a}}_{1}∼{\dot{a}}_{2}}x^{\sqrt{\frac{\left(v_{{\dot{a}}_{1}}+v_{{\dot{a}}_{2}}-2\right)}{\left(v_{{\dot{a}}_{1}}+v_{{\dot{a}}_{2}}\right)}}}\;\;ABS=\sum_{{\dot{a}}_{1}∼{\dot{a}}_{2}}\sqrt{\frac{\left(v_{{\dot{a}}_{1}}+v_{{\dot{a}}_{2}}-2\right)}{\left(v_{{\dot{a}}_{1}}+v_{{\dot{a}}_{2}}\right)}}$$

Shannon first popularized the concept of entropy in his 1948 work [17]. Entropy is the quantity of thermal energy per unit temperature in a system that is not accessible for meaningful work [18,19]. Because the work is derived from organized molecular motion, entropy is also a measure of a system’s molecular disorder or unpredictability [20,21]. In this article, we build the Niobium dioxide NbO${}_{2}$ and the metal–organic framework (MOF) to compute the *K*-Banhatti and redefined Zagreb entropies using *K*-Banhatti indices [22,23,24], and redefined Zagreb indices, respectively. The idea of entropy is extracted from Shazia Manzoor’s paper [25].

## 2. Valency-Based Entropy

The idea of edge-weighted graph entropy was introduced in 2009 [26], $G=(\left(V_{G},E_{G}\right),\phi \left(v_{{\dot{a}}_{1}}v_{{\dot{a}}_{2}}\right))$ for an edge-weighted graph, where $V_{G}$ is the vertex set, $E_{G}$ the edge set, and the edge-weight of an edge $\left(v_{{\dot{a}}_{1}}v_{{\dot{a}}_{2}}\right)$ is represented by $\phi (v_{{\dot{a}}_{1}}v_{{\dot{a}}_{2}})$. The entropy of an edge-weighted graph is defined as
(9)$$ENT_{\phi (G)}=-\sum_{{\dot{a}}_{1}∼{\dot{a}}_{2}}\frac{\phi (v_{{\dot{a}}_{1}}v_{{\dot{a}}_{2}})}{\sum_{{\dot{a}}_{1}∼{\dot{a}}_{2}}\phi \left(v_{{\dot{a}}_{1}}v_{{\dot{a}}_{2}}\right)}\log\{\frac{\phi (v_{{\dot{a}}_{1}}v_{{\dot{a}}_{2}})}{\sum_{{\dot{a}}_{1}∼{\dot{a}}_{2}}\phi \left(v_{{\dot{a}}_{1}}v_{{\dot{a}}_{2}}\right)}\}.$$


**The first K-Banhatti entropy**
Let $\phi \left(v_{{\dot{a}}_{1}}v_{{\dot{a}}_{2}}\right)=v_{{\dot{a}}_{1}}+v_{{\dot{a}}_{2}}$. Then, the first *K*-Banhatti index (Equation 1) is given by
$$\begin{array}{ccc} B_{1}\left(G\right) & = & \sum_{{\dot{a}}_{1}∼{\dot{a}}_{2}}\{v_{{\dot{a}}_{1}}+v_{{\dot{a}}_{2}}\}=\sum_{{\dot{a}}_{1}∼{\dot{a}}_{2}}\phi \left(v_{{\dot{a}}_{1}}v_{{\dot{a}}_{2}}\right). \end{array}$$Now, by inserting these values into Equation (Equation 9), the first *K*-Banhatti entropy is
(10)$$ENT_{B_{1}\left(G\right)}=\log\left(B_{1}\left(G\right)\right)-\frac{1}{B_{1}\left(G\right)}\log\{{\prod_{{\dot{a}}_{1}∼{\dot{a}}_{2}}\left[v_{{\dot{a}}_{1}}+v_{{\dot{a}}_{2}}\right]}^{\left[v_{{\dot{a}}_{1}}+v_{{\dot{a}}_{2}}\right]}\}.$$
**The second K-Banhatti entropy**
Let $\phi \left(v_{{\dot{a}}_{1}}v_{{\dot{a}}_{2}}\right)=v_{{\dot{a}}_{1}}\times v_{{\dot{a}}_{2}}$. Then, the second *K*-Banhatti index (Equation 2) is given by
$$\begin{array}{c} B_{2}\left(G\right)=\sum_{{\dot{a}}_{1}∼{\dot{a}}_{2}}\{\left(v_{{\dot{a}}_{1}}\times v_{{\dot{a}}_{2}}\right)\}=\sum_{{\dot{a}}_{1}∼{\dot{a}}_{2}}\phi \left(v_{{\dot{a}}_{1}}v_{{\dot{a}}_{2}}\right). \end{array}$$Now, by inserting these values into Equation (Equation 9), the second *K*-Banhatti entropy is
(11)$$ENT_{B_{2}\left(G\right)}=\log\left(B_{2}\left(G\right)\right)-\frac{1}{B_{2}\left(G\right)}\log\{{\prod_{{\dot{a}}_{1}∼{\dot{a}}_{2}}\left[v_{{\dot{a}}_{1}}\times v_{{\dot{a}}_{2}}\right]}^{\left[v_{{\dot{a}}_{1}}\times v_{{\dot{a}}_{2}}\right]}\}.$$
**The first K-hyper Banhatti entropy**
Let $\phi \left(v_{{\dot{a}}_{1}}v_{{\dot{a}}_{2}}\right)={\left(v_{{\dot{a}}_{1}}+v_{{\dot{a}}_{2}}\right)}^{2}$. Then, the first *K*-hyper Banhatti index (Equation 3) is given by
$$\begin{array}{c} HB_{1}\left(G\right)=\sum_{{\dot{a}}_{1}∼{\dot{a}}_{2}}\{{\left(v_{{\dot{a}}_{1}}+v_{{\dot{a}}_{2}}\right)}^{2}\}=\sum_{{\dot{a}}_{1}∼{\dot{a}}_{2}}\phi \left(v_{{\dot{a}}_{1}}v_{{\dot{a}}_{2}}\right). \end{array}$$Now, by inserting these values into Equation (Equation 9), the first *K*-hyper Banhatti entropy is
(12)$$ENT_{HB_{1}\left(G\right)}=\log\left(HB_{1}\left(G\right)\right)-\frac{1}{HB_{1}\left(G\right)}\log\{{\prod_{{\dot{a}}_{1}∼{\dot{a}}_{2}}\left[v_{{\dot{a}}_{1}}+v_{{\dot{a}}_{2}}\right]}^{2{\left[v_{{\dot{a}}_{1}}+v_{{\dot{a}}_{2}}\right]}^{2}}\}.$$
**The second K-hyper Banhatti entropy**
Let $\phi \left(v_{{\dot{a}}_{1}}v_{{\dot{a}}_{2}}\right)={\left(v_{{\dot{a}}_{1}}\times v_{{\dot{a}}_{2}}\right)}^{2}$. Then the second *K*-hyper Banhatti index (Equation 4) is given by
$$\begin{array}{ccc} HB_{2}\left(G\right) & = & \sum_{{\dot{a}}_{1}∼{\dot{a}}_{2}}\{{\left(v_{{\dot{a}}_{1}}\times v_{{\dot{a}}_{2}}\right)}^{2}\}=\sum_{{\dot{a}}_{1}∼{\dot{a}}_{2}}\phi \left(v_{{\dot{a}}_{1}}v_{{\dot{a}}_{2}}\right). \end{array}$$Now, by inserting these values into Equation (Equation 9), the second *K*-hyper Banhatti entropy is
(13)$$ENT_{HB_{2}\left(G\right)}=\log\left(HB_{1}\left(G\right)\right)-\frac{1}{HB_{1}\left(G\right)}\log\{{\prod_{{\dot{a}}_{1}∼{\dot{a}}_{2}}\left[v_{{\dot{a}}_{1}}\times v_{{\dot{a}}_{2}}\right]}^{2{\left[v_{{\dot{a}}_{1}}\times v_{{\dot{a}}_{2}}\right]}^{2}}\}.$$
**The first redefined Zagreb entropy**
Let $\phi \left(v_{{\dot{a}}_{1}}v_{{\dot{a}}_{2}}\right)=\frac{v_{{\dot{a}}_{1}}+v_{{\dot{a}}_{2}}}{v_{{\dot{a}}_{1}}v_{{\dot{a}}_{2}}}$. Then, the first redefined Zagreb index (Equation 5) is given by
$$\begin{array}{ccc} ReZG_{1} & = & \sum_{{\dot{a}}_{1}∼{\dot{a}}_{2}}\{\frac{v_{{\dot{a}}_{1}}+v_{{\dot{a}}_{2}}}{v_{{\dot{a}}_{1}}v_{{\dot{a}}_{2}}}\}=\sum_{{\dot{a}}_{1}∼{\dot{a}}_{2}}\phi \left(v_{{\dot{a}}_{1}}v_{{\dot{a}}_{2}}\right). \end{array}$$Now, by inserting these values into Equation (Equation 9), the first redefined Zagreb entropy is
(14)$$ENT_{ReZG_{1}}=\log\left(ReZG_{1}\right)-\frac{1}{ReZG_{1}}\log\{{\prod_{{\dot{a}}_{1}∼{\dot{a}}_{2}}\left[\frac{v_{{\dot{a}}_{1}}+v_{{\dot{a}}_{2}}}{v_{{\dot{a}}_{1}}v_{{\dot{a}}_{2}}}\right]}^{\left[\frac{v_{{\dot{a}}_{1}}+v_{{\dot{a}}_{2}}}{v_{{\dot{a}}_{1}}v_{{\dot{a}}_{2}}}\right]}\}.$$
**The second redefined Zagreb entropy**
Let $\phi \left(v_{{\dot{a}}_{1}}v_{{\dot{a}}_{2}}\right)=\frac{v_{{\dot{a}}_{1}}d_{v}}{v_{{\dot{a}}_{1}}+v_{{\dot{a}}_{2}}}$. Then, the second redefined index (Equation 6) is given by
$$\begin{array}{ccc} ReZG_{2} & = & \sum_{{\dot{a}}_{1}∼{\dot{a}}_{2}}\{\frac{v_{{\dot{a}}_{1}}v_{{\dot{a}}_{2}}}{v_{{\dot{a}}_{1}}+v_{{\dot{a}}_{2}}}\}=\sum_{{\dot{a}}_{1}∼{\dot{a}}_{2}}\phi \left(v_{{\dot{a}}_{1}}v_{{\dot{a}}_{2}}\right). \end{array}$$Now, by inserting these values into Equation (Equation 9), the second redefined Zagreb entropy is
(15)$$ENT_{ReZG_{2}}=\log\left(ReZG_{2}\right)-\frac{1}{ReZG_{2}}\log\{{\prod_{{\dot{a}}_{1}∼{\dot{a}}_{2}}\left[\frac{v_{{\dot{a}}_{1}}d_{v}}{v_{{\dot{a}}_{1}}+v_{{\dot{a}}_{2}}}\right]}^{\left[\frac{v_{{\dot{a}}_{1}}v_{{\dot{a}}_{2}}}{v_{{\dot{a}}_{1}}+v_{{\dot{a}}_{2}}}\right]}\}.$$
**The third redefined Zagreb entropy**
Let $\phi \left(v_{{\dot{a}}_{1}}v_{{\dot{a}}_{2}}\right)=\{\left(v_{{\dot{a}}_{1}}v_{{\dot{a}}_{2}}\right)\left(v_{{\dot{a}}_{1}}+v_{{\dot{a}}_{2}}\right)\}$. Then, the third redefined Zagreb index (Equation 7) is given by
$$\begin{array}{ccc} ReZG_{3} & = & \sum_{{\dot{a}}_{1}∼{\dot{a}}_{2}}\{\left(v_{{\dot{a}}_{1}}v_{{\dot{a}}_{2}}\right)\left(d_{u}+d_{v}\right)\}=\sum_{{\dot{a}}_{1}∼{\dot{a}}_{2}}\phi \left(v_{{\dot{a}}_{1}}v_{{\dot{a}}_{2}}\right). \end{array}$$Now, by inserting these values into Equation (Equation 9), the third redefined Zagreb entropy is
(16)$$ENT_{ReZG_{3}}=\log\left(ReZG_{3}\right)-\frac{1}{ReZG_{3}}\log\{{\prod_{{\dot{a}}_{1}∼{\dot{a}}_{2}}\left[\left(v_{{\dot{a}}_{1}}v_{{\dot{a}}_{2}}\right)\left(v_{{\dot{a}}_{1}}+v_{{\dot{a}}_{2}}\right)\right]}^{\left[\left(v_{{\dot{a}}_{1}}v_{{\dot{a}}_{2}}\right)\left(v_{{\dot{a}}_{1}}+v_{{\dot{a}}_{2}}\right)\right]}\}.$$
**Atom-bond sum connectivity entropy**
Let $\phi \left({\dot{a}}_{1}{\dot{a}}_{2}\right)=\{\sqrt{\frac{v_{{\dot{a}}_{1}}+v_{{\dot{a}}_{2}}-2}{v_{{\dot{a}}_{1}}+v_{{\dot{a}}_{2}}}}\}$. Then, the fourth atom-bond connectivity index (Equation 8) is given by
$$\begin{array}{ccc} ABS(G) & = & \sum_{{\dot{a}}_{1},{\dot{a}}_{2}\in E_{G}}^{}\{\sqrt{\frac{v_{{\dot{a}}_{1}}+v_{{\dot{a}}_{2}}-2}{v_{{\dot{a}}_{1}}+v_{{\dot{a}}_{2}}}}\}=\sum_{{\dot{a}}_{1},{\dot{a}}_{2}\in E_{G}}^{}\phi \left({\dot{a}}_{1}{\dot{a}}_{2}\right). \end{array}$$By inserting the values of ABS(G) into Equation (Equation 9), the atom-bond sum connectivity $\left(ENT_{ABC(G)}\right)$ entropy is
(17)$$ENT_{ABS(G)}=\log\left(ABS(G)\right)-\frac{1}{ABS(G)}\log\{{\prod_{{\dot{a}}_{1},{\dot{a}}_{2}\in E_{G}}\left(\sqrt{\frac{v_{{\dot{a}}_{1}}+v_{{\dot{a}}_{2}}-2}{v_{{\dot{a}}_{1}}+v_{{\dot{a}}_{2}}}}\right)}^{\left(\sqrt{\frac{v_{{\dot{a}}_{1}}+v_{{\dot{a}}_{2}}-2}{v_{{\dot{a}}_{1}}+v_{{\dot{a}}_{2}}}}\right)}\}.$$

## 3. Niobium Dioxide NbO${}_{2}$

Niobium Nb, a refractory metal, is a good choice for the initial shell of nuclear fusion reactors. It does, however, have a strong attraction for O${}_{2}$ and C, both of which are available in pyrotechnics and refrigerant-like liquids. As part of the first barrier, Nb is well known for its ability to interact very effectively with O${}_{2}$ [27]. As a result, reliable thermodynamic data on NbO, NbO${}_{2}$, Nb${}_{2}$O${}_{5}$, and other intermediate phases, such as Nb${}_{12}$O${}_{29}$, are very effective. In transistors, niobium monoxide is used as a gate electrode, and a (NbO/NbO${}_{2}$) junction may be used in robust switching devices. In this article, we will attempt to explain NbO${}_{2}$, which has a total atom count of 2+5s+5t+9st; see Figure 1.

There are three types of atoms in NbO${}_{2}$ based on their valency: eight atoms with valency 2, 8s+8t+4st−8 atoms with valency 3, and 2−3s−3t+5st atoms with valency 4. Table 1 shows the atom-bond partitions of NbO${}_{2}$ derived from these results.


**The first K-Banhatti entropy of NbO${}_{2}$**
Let NbO${}_{2}$ be a network of a niobium dioxide molecule. Then, by using Equation (Equation 1) and Table 1, the first *K*-Banhatti polynomial is
(18)$$\begin{array}{ccc} B_{1}\left({\mathrm{NbO}}_{2},x\right) & = & \sum_{E_{\left(2∼3\right)}}x^{2+3}+\sum_{E_{\left(3∼3\right)}}x^{3+3}+\sum_{E_{\left(3∼4\right)}}x^{3+4}+\sum_{E_{\left(4∼4\right)}}x^{4+4}\;\\ \; & = & 16x^{5}+8\left(2s+2t-3\right)x^{6}+4\left(3st-2s-2t+2\right)x^{7}\;\\ \; & + & 2\left(2st-s-t\right)x^{8}. \end{array}$$After simplifying Equation (Equation 18), we obtain the first *K*-Banhatti index by taking the first derivative at x=1.
(19)$$B_{1}\left({\mathrm{NbO}}_{2}\right)=116st+24s+24t-8.$$Now, we compute the first *K*-Banhatti entropy of NbO${}_{2}$ by using Table 1 and Equation (Equation 19) in Equation (Equation 10) in the following way:
$$\begin{array}{ccc} ENT_{B_{1}}\left({\mathrm{NbO}}_{2}\right) & = & \log\left(B_{1}\right)-\frac{1}{B_{1}}\log\{{\prod_{E_{\left(2,3\right)}}\left(v_{{\dot{a}}_{1}}+v_{{\dot{a}}_{2}}\right)}^{\left(v_{{\dot{a}}_{1}}+v_{{\dot{a}}_{2}}\right)}\times {\prod_{E_{\left(3,3\right)}}\left(v_{{\dot{a}}_{1}}+v_{{\dot{a}}_{2}}\right)}^{\left(v_{{\dot{a}}_{1}}+v_{{\dot{a}}_{2}}\right)}\;\\ & \times & {\prod_{E_{\left(3,4\right)}}\left(v_{{\dot{a}}_{1}}+v_{{\dot{a}}_{2}}\right)}^{\left(v_{{\dot{a}}_{1}}+v_{{\dot{a}}_{2}}\right)}\times {\prod_{E_{\left(4,4\right)}}\left(v_{{\dot{a}}_{1}}+v_{{\dot{a}}_{2}}\right)}^{\left(v_{{\dot{a}}_{1}}+v_{{\dot{a}}_{2}}\right)}\;\\ & = & \log\left(116st+24s+24t-8\right)-\frac{1}{116st+24s+24t-8}\log\{16{\left(4\right)}^{4}\;\\ & \times & 8\left(2s+2t-3\right){\left(5\right)}^{5}\times 4\left(3st-2s-2t+2\right){\left(6\right)}^{6}\times 2\left(2st-s-t\right){\left(8\right)}^{8}. \end{array}$$
**The second K-Banhatti entropy of NbO${}_{2}$**
Let NbO${}_{2}$ be a network of a niobium dioxide molecule. Then, by using Equation (Equation 2) and Table 1, the second *K*-Banhatti polynomial is
(20)$$\begin{array}{ccc} B_{2}\left({\mathrm{NbO}}_{2}\right) & = & \sum_{E_{\left(2∼3\right)}}x^{2\times 3}+\sum_{E_{\left(3∼3\right)}}x^{3\times 3}+\sum_{E_{\left(3∼4\right)}}x^{3\times 4}+\sum_{E_{\left(4∼4\right)}}x^{4\times 4}\;\\ \; & = & 16x^{6}+8\left(2s+2t-3\right)x^{9}+4\left(3st-2s-2t+2\right)x^{12}\;\\ \; & + & 2\left(2st-s-t\right)x^{16}. \end{array}$$Taking the first derivative of Equation (Equation 20) at x=1, we obtain the second *K*-Banhatti index
(21)$$B_{2}\left({\mathrm{NbO}}_{2}\right)=208st+16s+16t-24.$$Now, we compute the second *K*-Banhatti entropy of NbO${}_{2}$ by using Table 1 and Equation (Equation 21) in Equation (Equation 11) in the following way:
$$\begin{array}{ccc} ENT_{B_{2}}\left({\mathrm{NbO}}_{2}\right) & = & \log\left(B_{2}\right)-\frac{1}{B_{2}}\log\{{\prod_{E_{\left(2,3\right)}}\left(v_{{\dot{a}}_{1}}\times v_{{\dot{a}}_{2}}\right)}^{\left(v_{{\dot{a}}_{1}}\times v_{{\dot{a}}_{2}}\right)}\times {\prod_{E_{\left(3,3\right)}}\left(v_{{\dot{a}}_{1}}\times v_{{\dot{a}}_{2}}\right)}^{\left(v_{{\dot{a}}_{1}}\times v_{{\dot{a}}_{2}}\right)}\;\\ \; & \times & {\prod_{E_{\left(3,4\right)}}\left(v_{{\dot{a}}_{1}}\times v_{{\dot{a}}_{2}}\right)}^{\left(v_{{\dot{a}}_{1}}\times v_{{\dot{a}}_{2}}\right)}\times {\prod_{E_{\left(4,4\right)}}\left(v_{{\dot{a}}_{1}}\times v_{{\dot{a}}_{2}}\right)}^{\left(v_{{\dot{a}}_{1}}\times v_{{\dot{a}}_{2}}\right)}\}\;\\ \; & = & \log\left(208st+16s+16t-24\right)-\frac{1}{208st+16s+16t-24}\log\{16\left(6^{6}\right)\;\\ \; & \times & 8\left(2s+2t-3\right)9^{9}\times 4\left(3st-2s-2t+2\right)12^{12}\times 2\left(2st-s-t\right)16^{16}\}. \end{array}$$
**The first K-hyper Banhatti entropy of NbO${}_{2}$**
Let NbO${}_{2}$ be a network of a niobium dioxide molecule. Then, by using Equation (Equation 3) and Table 1, the first *K*-hyper Banhatti polynomial is
(22)$$\begin{array}{ccc} HB_{1}\left({\mathrm{NbO}}_{2}\right) & = & \sum_{E_{\left(2∼3\right)}}x^{{\left(2+3\right)}^{2}}+\sum_{E_{\left(3∼3\right)}}x^{{\left(3+3\right)}^{2}}+\sum_{E_{\left(3∼4\right)}}x^{{\left(3+4\right)}^{2}}+\sum_{E_{\left(4∼4\right)}}x^{{\left(4+4\right)}^{2}}\;\\ \; & = & 16x^{25}+8\left(2s+2t-3\right)x^{36}+4\left(3st-2s-2t+2\right)x^{49}\;\\ \; & + & 2\left(2st-s-t\right)x^{64}. \end{array}$$Taking the first derivative of Equation (Equation 22) at x=1, we obtain the first *K*-hyper Banhatti index
(23)$$HB_{1}\left({\mathrm{NbO}}_{2}\right)=844st+56s+56t-72.$$Now, we compute the first *K*-hyper Banhatti entropy of NbO${}_{2}$ by using Table 1 and Equation (Equation 23) in Equation (Equation 13) in the following way:
$$\begin{array}{ccc} ENT_{HB_{1}}\left({\mathrm{NbO}}_{2}\right) & = & \log\left(HB_{1}\right)-\frac{1}{HB_{1}}\log\{{\prod_{E_{\left(2,3\right)}}\left(v_{{\dot{a}}_{1}}+v_{{\dot{a}}_{2}}\right)}^{2{\left(v_{{\dot{a}}_{1}}+v_{{\dot{a}}_{2}}\right)}^{2}}\times {\prod_{E_{\left(3,3\right)}}\left(v_{{\dot{a}}_{1}}+v_{{\dot{a}}_{2}}\right)}^{2{\left(v_{{\dot{a}}_{1}}+v_{{\dot{a}}_{2}}\right)}^{2}}\;\\ \; & \times & {\prod_{E_{\left(3,4\right)}}\left(v_{{\dot{a}}_{1}}+v_{{\dot{a}}_{2}}\right)}^{2{\left(v_{{\dot{a}}_{1}}+v_{{\dot{a}}_{2}}\right)}^{2}}\times {\prod_{E_{\left(4,4\right)}}\left(v_{{\dot{a}}_{1}}+v_{{\dot{a}}_{2}}\right)}^{2{\left(v_{{\dot{a}}_{1}}+v_{{\dot{a}}_{2}}\right)}^{2}}\;\\ \; & = & \log\left(944st+136s+200t\right)-\frac{1}{944st+136s+200t}\log\{16\left(5^{50}\right)\;\\ \; & \times & 8\left(2s+2t-3\right)\left(6^{72}\right)\times 4\left(3st-2s-2t+2\right)\left(7^{98}\right)\times 2\left(2st-s-t\right)\left(8^{128}\right). \end{array}$$
**The second K-hyper Banhatti entropy of NbO${}_{2}$**
Let NbO${}_{2}$ be a network of a niobium dioxide molecule. Then, by using Equation (Equation 4) and Table 1, the second *K*-hyper Banhatti polynomial is
(24)$$\begin{array}{ccc} HB_{2}\left({\mathrm{NbO}}_{2}\right) & = & \sum_{E_{\left(2∼3\right)}}x^{{\left(2\times 3\right)}^{2}}+\sum_{E_{\left(3∼3\right)}}x^{{\left(3\times 3\right)}^{2}}+\sum_{E_{\left(3∼4\right)}}x^{{\left(3\times 4\right)}^{2}}+\sum_{E_{\left(4∼4\right)}}x^{{\left(4\times 4\right)}^{2}}\;\\ \; & = & 16x^{36}+8\left(2s+2t-3\right)x^{81}+4\left(3st-2s-2t+2\right)x^{144}\;\\ \; & + & 2\left(2st-s-t\right)x^{256}. \end{array}$$Taking the first derivative of Equation (Equation 24) at x=1, we obtain the second *K*-hyper Banhatti index
(25)$$HB_{2}\left({\mathrm{NbO}}_{2}\right)=2752st-368s-368t-216.$$Now, we compute the second *K*-hyper Banhatti entropy of NbO${}_{2}$ by using Table 1 and Equation (Equation 25) in Equation (Equation 13) in the following way:
$$\begin{array}{ccc} ENT_{HB_{1}}\left({\mathrm{NbO}}_{2}\right) & = & \log\left(HB_{1}\right)-\frac{1}{HB_{1}}\log\{{\prod_{E_{\left(2,3\right)}}\left(v_{{\dot{a}}_{1}}\times v_{{\dot{a}}_{2}}\right)}^{2{\left(v_{{\dot{a}}_{1}}\times v_{{\dot{a}}_{2}}\right)}^{2}}\times {\prod_{E_{\left(3,3\right)}}\left(v_{{\dot{a}}_{1}}\times v_{{\dot{a}}_{2}}\right)}^{2{\left(v_{{\dot{a}}_{1}}\times v_{{\dot{a}}_{2}}\right)}^{2}}\;\\ \; & \times & {\prod_{E_{\left(3,4\right)}}\left(v_{{\dot{a}}_{1}}\times v_{{\dot{a}}_{2}}\right)}^{2{\left(v_{{\dot{a}}_{1}}\times v_{{\dot{a}}_{2}}\right)}^{2}}\times {\prod_{E_{\left(4,4\right)}}\left(v_{{\dot{a}}_{1}}\times v_{{\dot{a}}_{2}}\right)}^{2{\left(v_{{\dot{a}}_{1}}\times v_{{\dot{a}}_{2}}\right)}^{2}}\;\\ \; & = & \log\left(2752st-368s-368t-216\right)-\frac{1}{2752st-368s-368t-216}\log\{16{\left(6\right)}^{72}\;\\ \; & \times & 8\left(2s+2t-3\right)9^{81}\times 4\left(3st-2s-2t+2\right)12^{288}\times 2\left(2st-s-t\right)16^{512}. \end{array}$$
**The first redefined Zagreb entropy of NbO${}_{2}$**
Let NbO${}_{2}$ be a network of a niobium dioxide molecule. Then, by using Equation (Equation 5) and Table 1, the first redefined Zagreb polynomial is
(26)$$\begin{array}{ccc} ReZG_{1}\left({\mathrm{NbO}}_{2}\right) & = & \sum_{E_{\left(2∼3\right)}}x^{\frac{2+3}{2\times 3}}+\sum_{E_{\left(3∼3\right)}}x^{\frac{3+3}{3\times 3}}+\sum_{E_{\left(3∼3\right)}}x^{\frac{3+4}{3\times 4}}+\sum_{E_{\left(4∼4\right)}}x^{\frac{4+4}{4\times 4}}\;\\ \; & = & 16x^{\frac{5}{6}}+8\left(2s+2t-3\right)x^{\frac{2}{3}}+4\left(3st-2s-2t+2\right)x^{\frac{7}{12}}\;\\ \; & + & 2\left(2st-s-t\right)x^{\frac{1}{2}}. \end{array}$$Taking the first derivative of Equation (Equation 26) at x=1, we obtain the first redefined Zagreb index
(27)$$ReZG_{1}\left({\mathrm{NbO}}_{2}\right)=9st+5s+5t+2.$$Now, we compute the first redefined Zagreb entropy by using Table 1 and Equation (Equation 27) in Equation (Equation 14) in the following way:
$$\begin{array}{ccc} ENT_{ReZG_{1}}\left({\mathrm{NbO}}_{2}\right) & = & \log\left(ReZG_{1}\right)-\frac{1}{ReZG_{1}}\log\{{\prod_{E_{\left(2,3\right)}}\left[\frac{v_{{\dot{a}}_{1}}+v_{{\dot{a}}_{2}}}{v_{{\dot{a}}_{1}}v_{{\dot{a}}_{2}}}\right]}^{\left[\frac{v_{{\dot{a}}_{1}}+v_{{\dot{a}}_{2}}}{v_{{\dot{a}}_{1}}d_{v}}\right]}\;\\ \; & \times & {\prod_{E_{\left(3,3\right)}}\left[\frac{v_{{\dot{a}}_{1}}+v_{{\dot{a}}_{2}}}{v_{{\dot{a}}_{1}}v_{{\dot{a}}_{2}}}\right]}^{\left[\frac{v_{{\dot{a}}_{1}}+d_{v}}{v_{{\dot{a}}_{1}}v_{{\dot{a}}_{2}}}\right]}\times {\prod_{E_{\left(3,4\right)}}\left[\frac{v_{{\dot{a}}_{1}}+v_{{\dot{a}}_{2}}}{v_{{\dot{a}}_{1}}v_{{\dot{a}}_{2}}}\right]}^{\left[\frac{v_{{\dot{a}}_{1}}+v_{{\dot{a}}_{2}}}{v_{{\dot{a}}_{1}}v_{{\dot{a}}_{2}}}\right]}\;\\ \; & \times & {\prod_{E_{\left(4,4\right)}}\left[\frac{v_{{\dot{a}}_{1}}+v_{{\dot{a}}_{2}}}{v_{{\dot{a}}_{1}}v_{{\dot{a}}_{2}}}\right]}^{\left[\frac{v_{{\dot{a}}_{1}}+d_{v}}{v_{{\dot{a}}_{1}}v_{{\dot{a}}_{2}}}\right]}\}\;\\ \; & = & \log8(9st+5s+5t+2)\;\\ \; & - & \frac{1}{8(9st+5s+5t+2)}\log\{16{\left(\frac{5}{6}\right)}^{\frac{5}{6}}\;\\ \; & \times & 8\left(2s+2t-3\right){\left(\frac{2}{3}\right)}^{\frac{2}{3}}\;\\ \; & \times & 4\left(3st-2s-2t+2\right){\left(\frac{7}{12}\right)}^{\frac{7}{12}}\times 2\left(2st-s-t\right){\left(\frac{8}{16}\right)}^{\frac{8}{16}}\}. \end{array}$$
**The second redefined Zagreb entropy of NbO${}_{2}$**
Let NbO${}_{2}$ be a network of a niobium dioxide molecule. Then, by using Equation (Equation 6) and Table 1, the second redefined Zagreb polynomial is
(28)$$\begin{array}{ccc} ReZG_{2}\left({\mathrm{NbO}}_{2}\right) & = & \sum_{E_{\left(2∼3\right)}}x^{\frac{2\times 3}{2+3}}+\sum_{E_{\left(3∼3\right)}}x^{\frac{3\times 3}{3+3}}+\sum_{E_{\left(3∼4\right)}}x^{\frac{3\times 4}{3+4}}+\sum_{E_{\left(4∼4\right)}}x^{\frac{4\times 4}{4+4}}\;\\ \; & = & 16x^{\frac{6}{5}}+8\left(2s+2t-3\right)x^{\frac{3}{2}}+4\left(3st-2s-2t+2\right)x^{\frac{12}{7}}\;\\ \; & + & 2\left(2st-s-t\right)x^{2}. \end{array}$$Taking the first derivative of Equation (Equation 28) at x=1, we obtain the second redefined Zagreb index
(29)$$ReZG_{2}\left({\mathrm{NbO}}_{2}\right)=\frac{4}{7}\left(25st+11s+11t-27\right).$$Now, we compute the second redefined Zagreb entropy by using Table 1 and Equation (Equation 29) in Equation (Equation 15) in the following way:
$$\begin{array}{ccc} ENT_{ReZG_{2}}\left({\mathrm{NbO}}_{2}\right)\; & = & \log\left(ReZG_{2}\right)-\frac{1}{ReZG_{2}}\log\{{\prod_{E_{\left(2,3\right)}}\left[\frac{v_{{\dot{a}}_{1}}v_{{\dot{a}}_{2}}}{v_{{\dot{a}}_{1}}+v_{{\dot{a}}_{2}}}\right]}^{\left[\frac{v_{{\dot{a}}_{1}}v_{{\dot{a}}_{2}}}{v_{{\dot{a}}_{1}}+v_{{\dot{a}}_{2}}}\right]}\;\\ \times {\prod_{E_{\left(3,3\right)}}\left[\frac{v_{{\dot{a}}_{1}}v_{{\dot{a}}_{2}}}{v_{{\dot{a}}_{1}}+v_{{\dot{a}}_{2}}}\right]}^{\left[\frac{v_{{\dot{a}}_{1}}v_{{\dot{a}}_{2}}}{d_{u}+v_{{\dot{a}}_{2}}}\right]}\; & \times & {\prod_{E_{\left(3,4\right)}}\left[\frac{v_{{\dot{a}}_{1}}v_{{\dot{a}}_{2}}}{v_{{\dot{a}}_{1}}+v_{{\dot{a}}_{2}}}\right]}^{\left[\frac{v_{{\dot{a}}_{1}}v_{{\dot{a}}_{2}}}{v_{{\dot{a}}_{1}}+v_{{\dot{a}}_{2}}}\right]}\times {\prod_{E_{\left(4,4\right)}}\left[\frac{v_{{\dot{a}}_{1}}v_{{\dot{a}}_{2}}}{v_{{\dot{a}}_{1}}+v_{{\dot{a}}_{2}}}\right]}^{\left[\frac{v_{{\dot{a}}_{1}}v_{{\dot{a}}_{2}}}{v_{{\dot{a}}_{1}}+v_{{\dot{a}}_{2}}}\right]}\}\;\\ \; & = & \log\frac{4}{7}\left(25st+11s+11t-27\right)\;\\ \; & - & \frac{7}{4(25st+11s+11t-27)}\log\{16{\left(\frac{6}{5}\right)}^{\frac{6}{5}}\;\\ \; & \times & 8\left(2s+2t-3\right){\left(\frac{9}{6}\right)}^{\frac{9}{6}}\times 4\left(3st-2s-2t+2\right){\left(\frac{12}{7}\right)}^{\frac{12}{7}}\;\\ \; & \times & 2\left(2st-s-t\right){\left(\frac{16}{8}\right)}^{\frac{16}{8}}\}. \end{array}$$
**The third redefined Zagreb entropy of NbO${}_{2}$**
Let NbO${}_{2}$ be a network of a niobium dioxide molecule. Then, by using Equation (Equation 7) and Table 1, the third redefined Zagreb polynomial is
(30)$$\begin{array}{ccc} ReZG_{3}\left({\mathrm{NbO}}_{2}\right) & = & \sum_{E_{\left(2∼3\right)}}x^{\left(2\times 3)(2+3\right)}+\sum_{E_{\left(3∼3\right)}}x^{\left(3\times 3)(3+3\right)}+\sum_{E_{\left(3∼4\right)}}x^{\left(3\times 4)(3+4\right)}+\sum_{E_{\left(4∼4\right)}}x^{\left(4\times 4)(4+4\right)}\;\\ \; & = & 16x^{30}+8\left(2s+2t-3\right)x^{54}+4\left(3st-2s-2t+2\right)x^{84}\;\\ \; & + & 2\left(2st-s-t\right)x^{128}. \end{array}$$Taking the first derivative of Equation (Equation 30) at x=1, we obtain the third redefined Zagreb index
(31)$$ReZG_{3}\left({\mathrm{NbO}}_{2}\right)=8\left(95st-4s-4t-9\right).$$Now, we compute the third redefined Zagreb entropy by using Table 1 and Equation (Equation 31) in Equation (Equation 16) in the following way:
$$\begin{array}{ccc} ENT_{ReZG_{3}}\left({\mathrm{NbO}}_{2}\right) & = & \log\left(ReZG_{3}\right)-\frac{1}{ReZG_{3}}\log\{{\prod_{E_{\left(2,3\right)}}\left[\left(d_{u}v_{{\dot{a}}_{2}}\right)\left(d_{u}+v_{{\dot{a}}_{2}}\right)\right]}^{\left[\left(v_{{\dot{a}}_{1}}v_{{\dot{a}}_{2}}\right)\left(v_{{\dot{a}}_{1}}+v_{{\dot{a}}_{2}}\right)\right]}\;\\ \; & \times & {\prod_{E_{\left(3,3\right)}}\left[\left(v_{{\dot{a}}_{1}}v_{{\dot{a}}_{2}}\right)\left(v_{{\dot{a}}_{1}}+v_{{\dot{a}}_{2}}\right)\right]}^{\left[\left(d_{u}v_{{\dot{a}}_{2}}\right)\left(v_{{\dot{a}}_{1}}+v_{{\dot{a}}_{2}}\right)\right]}\;\\ \; & \times & {\prod_{E_{\left(3,4\right)}}\left[\left(v_{{\dot{a}}_{1}}v_{{\dot{a}}_{2}}\right)\left(v_{{\dot{a}}_{1}}+v_{{\dot{a}}_{2}}\right)\right]}^{\left[\left(v_{{\dot{a}}_{1}}v_{{\dot{a}}_{2}}\right)\left(v_{{\dot{a}}_{1}}+v_{{\dot{a}}_{2}}\right)\right]}\;\\ \; & \times & {\prod_{E_{\left(4,4\right)}}\left[\left(v_{{\dot{a}}_{1}}v_{{\dot{a}}_{2}}\right)\left(v_{{\dot{a}}_{1}}+v_{{\dot{a}}_{2}}\right)\right]}^{\left[\left(v_{{\dot{a}}_{1}}v_{{\dot{a}}_{2}}\right)\left(v_{{\dot{a}}_{1}}+v_{{\dot{a}}_{2}}\right)\right]}\}\;\\ \; & = & \log8(95st-4s-4t-9)-\frac{1}{8(95st-4s-4t-9)}\log\{16{\left(30\right)}^{30}\;\\ \; & \times & 8\left(2s+2t-3\right)54^{54}\times 4\left(3st-2s-2t+2\right)84^{84}\;\\ \; & \times & 2\left(2st-s-t\right)128^{128}\}. \end{array}$$
**Atom-bond sum connectivity entropy of NbO${}_{2}$**
Let NbO${}_{2}$ be a network of a niobium dioxide molecule. Then, using Equation (Equation 8) and Table 1, the atom-bond sum connectivity polynomial is
(32)$$\begin{array}{ccc} ABS({\mathrm{NbO}}_{2}) & = & \sum_{E_{\left(2∼3\right)}}x^{\sqrt{\frac{2+3-2}{2+3}}}+\sum_{E_{\left(3∼3\right)}}x^{\sqrt{\frac{3+3-2}{3+3}}}+\sum_{E_{\left(3∼4\right)}}x^{\sqrt{\frac{4+3-2}{4+3}}}+\sum_{E_{\left(4∼4\right)}}x^{\sqrt{\frac{4+4-2}{4+4}}}\;\\ & = & 16x^{\sqrt{\frac{3}{5}}}+8\left(2s+2t-3\right)x^{\frac{2}{\sqrt{6}}}+4\left(3st-2s-2t+2\right)x^{\sqrt{\frac{5}{7}}}\;\\ & + & 2\left(2st-s-t\right)x^{\frac{\sqrt{7}}{2}}. \end{array}$$Taking the first derivative of Equation (Equation 32) at x=1, we obtain the atom-bond sum connectivity index
(33)$$ABS\left({\mathrm{NbO}}_{2}\right)=16\sqrt{\frac{3}{5}}+8\left(2s+2t-3\right)\frac{2}{\sqrt{6}}+4\left(3st-2s-2t+2\right)\sqrt{\frac{5}{7}}+2\left(2st-s-t\right)\frac{\sqrt{7}}{2}.$$Now, we compute the atom-bond sum connectivity entropy by using Table 1 and Equation (Equation 33) in Equation (Equation 17) in the following way:
$$\begin{array}{ccc} ENT_{ABS}\left({\mathrm{NbO}}_{2}\right) & = & \log\left(ABS\right)-\frac{1}{ABS}\log\{{\prod_{E_{\left(2,3\right)}}\left[\sqrt{\frac{\left(v_{{\dot{a}}_{1}}+v_{{\dot{a}}_{2}}-2\right)}{\left(v_{{\dot{a}}_{1}}+v_{{\dot{a}}_{2}}\right)}}\right]}^{\left[\sqrt{\frac{\left(v_{{\dot{a}}_{1}}+v_{{\dot{a}}_{2}}-2\right)}{\left(v_{{\dot{a}}_{1}}+v_{{\dot{a}}_{2}}\right)}}\right]}\;\\ \; & \times & {\prod_{E_{\left(3,3\right)}}\left[\sqrt{\frac{\left(v_{{\dot{a}}_{1}}+v_{{\dot{a}}_{2}}-2\right)}{\left(v_{{\dot{a}}_{1}}+v_{{\dot{a}}_{2}}\right)}}\right]}^{\left[\sqrt{\frac{\left(v_{{\dot{a}}_{1}}+v_{{\dot{a}}_{2}}-2\right)}{\left(v_{{\dot{a}}_{1}}+v_{{\dot{a}}_{2}}\right)}}\right]}\times {\prod_{E_{\left(3,4\right)}}\left[\sqrt{\frac{\left(v_{{\dot{a}}_{1}}+v_{{\dot{a}}_{2}}-2\right)}{\left(v_{{\dot{a}}_{1}}+v_{{\dot{a}}_{2}}\right)}}\right]}^{\left[\sqrt{\frac{\left(v_{{\dot{a}}_{1}}+v_{{\dot{a}}_{2}}-2\right)}{\left(v_{{\dot{a}}_{1}}+v_{{\dot{a}}_{2}}\right)}}\right]}\;\\ \; & \times & {\prod_{E_{\left(4,4\right)}}\left[\sqrt{\frac{\left(v_{{\dot{a}}_{1}}+v_{{\dot{a}}_{2}}-2\right)}{\left(v_{{\dot{a}}_{1}}+v_{{\dot{a}}_{2}}\right)}}\right]}^{\left[\sqrt{\frac{\left(v_{{\dot{a}}_{1}}+v_{{\dot{a}}_{2}}-2\right)}{\left(v_{{\dot{a}}_{1}}+v_{{\dot{a}}_{2}}\right)}}\right]}\}\;\\ \; & = & \log\left(ABS\right)-\frac{1}{ABS}\log\{16{\left(\sqrt{\frac{3}{5}}\right)}^{\sqrt{\frac{3}{5}}}\times 8\left(2s+2t-3\right){\left(\sqrt{\frac{5}{6}}\right)}^{\sqrt{\frac{5}{6}}}\;\\ \; & \times & 4\left(3st-2s-2t+2\right){\left(\sqrt{\frac{5}{7}}\right)}^{\sqrt{\frac{5}{7}}}\times 2\left(2st-s-t\right){\left(\frac{\sqrt{7}}{2}\right)}^{\frac{\sqrt{7}}{2}}\}. \end{array}$$

### Comparison

In this section, we compare the *K*-Banhatti indices namely $B_{1}$ (first *K*-Banhatti index), $B_{2}$ (second *K*-Banhatti index), $HB_{1}$ (first hyper *K*-Banhatti index), $HB_{2}$ (second hyper *K*-Banhatti index) and the redefined Zagreb indices ($ReG_{1}$, $ReG_{2}$, $ReG_{3}$) for NbO${}_{2}$ numerically and graphically in Table 2 and Figure 2, respectively.

## 4. Metal–Organic Framework

Metal–organic frameworks are distinguished by their three-dimensional frameworks composed of metallic ions. This metal–organic framework has the molecular formula FeTPyP–Co, where Fe denotes iron, TPyP denotes tetrakis pyridyl porphyrin, and Co denotes cobalt [28]. All metal ions and organic molecules in the ${\mathrm{MOF}}_{\left(s,t\right)}$ network can accommodate a wide range of guest molecules. Metal–organic frameworks have several uses, including as energy storage devices, gas storage, heterogeneous catalysis, and chemical evaluation. We will examine a 2D structure of a metal–organic framework called ${\mathrm{MOF}}_{\left(s,t\right)}$, where *s* and *t* are the unit cells in a row and column, respectively. The ${\mathrm{MOF}}_{\left(2,2\right)}$ is shown in Figure 3. There are 74st atoms in the ${\mathrm{MOF}}_{\left(s,t\right)}$, and 2(44st−s−t)+1 atom-bonds are used, as Figure 3 of ${\mathrm{MOF}}_{\left(2,2\right)}$ demonstrates.

The atom-bonds partition of the ${\mathrm{MOF}}_{\left(s,t\right)}$ is shown in Table 3.
$$\begin{array}{ccc} E_{\left(1∼3\right)} & = & \{e=v_{{\dot{a}}_{1}}∼v_{{\dot{a}}_{2}},\forall \;{\dot{a}}_{1},{\dot{a}}_{2}\in E\left({\mathrm{MOF}}_{\left(s,t\right)}\right)|\left(v_{{\dot{a}}_{1}}\right)=1,\left(v_{{\dot{a}}_{2}}\right)=3\},\\ E_{\left(2∼3\right)} & = & \{e=v_{{\dot{a}}_{1}}∼v_{{\dot{a}}_{2}},\forall \;{\dot{a}}_{1},{\dot{a}}_{2}\in E\left({\mathrm{MOF}}_{\left(s,t\right)}\right)|\left(v_{{\dot{a}}_{1}}\right)=2,\left(v_{{\dot{a}}_{2}}\right)=3\},\\ E_{\left(3∼3\right)} & = & \{e=v_{{\dot{a}}_{1}}∼v_{{\dot{a}}_{2}},\forall \;{\dot{a}}_{1},{\dot{a}}_{2}\in E\left({\mathrm{MOF}}_{\left(s,t\right)}\right)|\left(v_{{\dot{a}}_{1}}\right)=3,\left(v_{{\dot{a}}_{2}}\right)=3\},\\ E_{\left(3∼4\right)} & = & \left\{e=v_{{\dot{a}}_{1}}∼v_{{\dot{a}}_{2}},\forall \;{\dot{a}}_{1},{\dot{a}}_{2}\in E\left({\mathrm{MOF}}_{\left(s,t\right)}\right)|\left(v_{{\dot{a}}_{1}}\right)=3,\left(v_{{\dot{a}}_{2}}\right)=4\right\} \end{array}$$


**The first K-Banhatti entropy of ${\mathrm{MOF}}_{\left(s,t\right)}$**
Let ${\mathrm{MOF}}_{\left(s,t\right)}$ be a metal–organic framework. Then, using Equation (Equation 1) and Table 3, the first *K*-Banhatti polynomial is
(34)$$\begin{array}{ccc} B_{1}\left({\mathrm{MOF}}_{\left(s,t\right)},x\right) & = & \sum_{E_{\left(1∼3\right)}}x^{1+3}+\sum_{E_{\left(2∼3\right)}}x^{2+3}+\sum_{E_{\left(3∼3\right)}}x^{3+3}+\sum_{E_{\left(3∼4\right)}}x^{3+4}\;\\ \; & = & \left(24st+1\right)x^{4}+6\left(s+t-1\right)x^{5}+2\left(28st-2s-2t+1\right)x^{6}\;\\ \; & + & 4\left(2st-s-t+1\right)x^{7}. \end{array}$$Taking the first derivative of Equation (Equation 34) at x=1, we obtain the first *K*-Banhatti index
(35)$$B_{1}\left({\mathrm{MOF}}_{\left(s,t\right)}\right)=2\left(244st-11s-11t+2\right).$$Now, we compute the 1*st*
*K*-Banhatti entropy of (${\mathrm{MOF}}_{\left(s,t\right)}$) by using Table 3 and Equation (Equation 35) in Equation (Equation 10) in the following way:
$$\begin{array}{ccc} ENT_{B_{1}}\left({\mathrm{MOF}}_{\left(s,t\right)},x\right) & = & \log\left(B_{1}\right)-\frac{1}{B_{1}}\log\{{\prod_{E_{\left(1,3\right)}}\left(v_{{\dot{a}}_{1}}+v_{{\dot{a}}_{2}}\right)}^{\left(v_{{\dot{a}}_{1}}+v_{{\dot{a}}_{2}}\right)}\times {\prod_{E_{\left(2,3\right)}}\left(v_{{\dot{a}}_{1}}+v_{{\dot{a}}_{2}}\right)}^{\left(v_{{\dot{a}}_{1}}+v_{{\dot{a}}_{2}}\right)}\\ & \times & {\prod_{E_{\left(3,3\right)}}\left(v_{{\dot{a}}_{1}}+v_{{\dot{a}}_{2}}\right)}^{\left(v_{{\dot{a}}_{1}}+v_{{\dot{a}}_{2}}\right)}\times {\prod_{E_{\left(3,4\right)}}\left(v_{{\dot{a}}_{1}}+v_{{\dot{a}}_{2}}\right)}^{\left(v_{{\dot{a}}_{1}}+v_{{\dot{a}}_{2}}\right)}. \end{array}$$After simplification, we obtain
(36)$$\begin{array}{ccc} ENT_{B_{1}}\left({\mathrm{MOF}}_{\left(s,t\right)},x\right) & = & \log2(244st-11s-11t+2)\;\\ \; & - & \frac{1}{2(244st-11s-11t+2)}\log\{\left(24st+1\right)4^{4}\;\\ \; & \times & 6\left(s+t-1\right)5^{5}\times 2\left(28st-2s-2t+1\right)6^{6}\;\\ \; & \times & 4\left(2st-s-t+1\right)7^{7}\}. \end{array}$$
**The second K-Banhatti entropy of ${\mathrm{MOF}}_{\left(s,t\right)}$**
Let ${\mathrm{MOF}}_{\left(s,t\right)}$ be a metal–organic framework. Then, using Equation (Equation 1) and Table 3, the second *K*-Banhatti polynomial is
(37)$$\begin{array}{ccc} B_{2}\left({\mathrm{MOF}}_{\left(s,t\right)},x\right) & = & \sum_{E_{\left(1∼3\right)}}x^{1\times 3}+\sum_{E_{\left(2∼3\right)}}x^{2\times 3}+\sum_{E_{\left(3∼3\right)}}x^{3\times 3}+\sum_{E_{\left(3∼4\right)}}x^{3\times 4}\;\\ & = & \left(24st+1\right)x^{3}+6\left(s+t-1\right)x^{6}+2\left(28st-2s-2t+1\right))x^{9}\;\\ & + & 4\left(2st-s-t+1\right)x^{12}. \end{array}$$Taking the first derivative of Equation (Equation 37) at x=1, we obtain the second *K*-Banhatti index
(38)$$B_{2}\left({\mathrm{MOF}}_{\left(s,t\right)}\right)=3\left(224st-16s-16t+11\right).$$Now, we compute the second *K*-Banhatti entropy of $\left({\mathrm{MOF}}_{\left(s,t\right)}\right)$ by using Table 3 and Equation (Equation 38) in Equation (Equation 11) in the following way:
$$\begin{array}{ccc} ENT_{B_{2}}\left({\mathrm{MOF}}_{\left(s,t\right)}\right) & = & \log\left(B_{2}\right)-\frac{1}{B_{2}}\log\{{\prod_{E_{\left(1,3\right)}}\left(v_{{\dot{a}}_{1}}\times v_{{\dot{a}}_{2}}\right)}^{\left(v_{{\dot{a}}_{1}}\times v_{{\dot{a}}_{2}}\right)}\times {\prod_{E_{\left(2,3\right)}}\left(v_{{\dot{a}}_{1}}\times v_{{\dot{a}}_{2}}\right)}^{\left(v_{{\dot{a}}_{1}}\times v_{{\dot{a}}_{2}}\right)}\\ & \times & {\prod_{E_{\left(3,3\right)}}\left(v_{{\dot{a}}_{1}}\times v_{{\dot{a}}_{2}}\right)}^{\left(v_{{\dot{a}}_{1}}\times v_{{\dot{a}}_{2}}\right)}\times {\prod_{E_{\left(3,4\right)}}\left(v_{{\dot{a}}_{1}}\times v_{{\dot{a}}_{2}}\right)}^{\left(v_{{\dot{a}}_{1}}\times v_{{\dot{a}}_{2}}\right)}\}\\ & = & \log\left(3(224st-16s-16t+11\right)\\ & - & \frac{1}{3(224st-16s-16t+11)}\log\{\left(24st+1\right)3^{3}\\ & \times & 6\left(s+t-1\right)6^{6}\times 2\left(28st-2s-2t+1\right)9^{9}\times 4\left(2st-s-t+1\right)12^{12}\}. \end{array}$$
**The first K-hyper Banhatti entropy of ${\mathrm{MOF}}_{\left(s,t\right)}$**
Let ${\mathrm{MOF}}_{\left(s,t\right)}$ be a metal–organic framework. Then, using Equation (Equation 3) and Table 3, the first *K*-hyper Banhatti polynomial is
(39)$$\begin{array}{ccc} HB_{1}\left({\mathrm{MOF}}_{\left(s,t\right)},x\right) & = & \sum_{E_{\left(1∼3\right)}}x^{{\left(1+3\right)}^{2}}+\sum_{E_{\left(2∼3\right)}}x^{{\left(2+3\right)}^{2}}+\sum_{E_{\left(3∼3\right)}}x^{{\left(3+3\right)}^{2}}+\sum_{E_{\left(3∼4\right)}}x^{{\left(3+4\right)}^{2}}\;\\ & = & \left(24st+1\right)x^{16}+6\left(s+t-1\right)x^{25}+2\left(28st-2s-2t+1\right)x^{36}\;\\ & + & 4\left(2st-s-t+1\right)x^{49}. \end{array}$$Taking the first derivative of Equation (Equation 39) at x=1, we obtain the first *K*-hyper Banhatti index
(40)$$HB_{1}\left({\mathrm{MOF}}_{\left(s,t\right)}\right)=2\left(1396st-95s-95t+67\right).$$Now, we compute the first *K*-hyper Banhatti entropy of ${\mathrm{MOF}}_{\left(s,t\right)}$ by using Table 3 and Equation (Equation 40) in Equation (Equation 12) in the following way:
$$\begin{array}{ccc} ENT_{HB_{1}}\left({\mathrm{MOF}}_{\left(s,t\right)},x\right) & = & \log\left(HB_{1}\right)-\frac{1}{HB_{1}}\log\{{\prod_{E_{\left(1,3\right)}}\left(v_{{\dot{a}}_{1}}+v_{{\dot{a}}_{2}}\right)}^{2{\left(v_{{\dot{a}}_{1}}+v_{{\dot{a}}_{2}}\right)}^{2}}\\ & \times & {\prod_{E_{\left(2,3\right)}}\left(v_{{\dot{a}}_{1}}+v_{{\dot{a}}_{2}}\right)}^{2{\left(v_{{\dot{a}}_{1}}+v_{{\dot{a}}_{2}}\right)}^{2}}\times {\prod_{E_{\left(3,3\right)}}\left(v_{{\dot{a}}_{1}}+v_{{\dot{a}}_{2}}\right)}^{2{\left(v_{{\dot{a}}_{1}}+v_{{\dot{a}}_{2}}\right)}^{2}}\\ & \times & {\prod_{E_{\left(4,4\right)}}\left(v_{{\dot{a}}_{1}}+v_{{\dot{a}}_{2}}\right)}^{2{\left(v_{{\dot{a}}_{1}}+v_{{\dot{a}}_{2}}\right)}^{2}}. \end{array}$$After simplification, we obtain
$$\begin{array}{ccc} & = & \log2(1396st-95s-95t+67)-\frac{1}{2(1396st-95s-95t+67)}\log\{\left(24st+1\right)4^{32}\\ & \times & 6\left(s+t-1\right)5^{50}\times 2\left(28st-2s-2t+1\right)6^{72}\times 4\left(2st-s-t+1\right)7^{98}. \end{array}$$
**The second K-hyper Banhatti entropy of ${\mathrm{MOF}}_{\left(s,t\right)}$**
Let ${\mathrm{MOF}}_{\left(s,t\right)}$ be a metal–organic framework. Then, by using Equation (Equation 4) and Table 3, the second *K*-Banhatti polynomial is
(41)$$\begin{array}{ccc} HB_{2}\left({\mathrm{MOF}}_{\left(s,t\right)},x\right) & = & \sum_{E_{\left(1∼3\right)}}x^{{\left(1\times 3\right)}^{2}}+\sum_{E_{\left(2∼3\right)}}x^{{\left(2\times 3\right)}^{2}}+\sum_{E_{\left(3∼3\right)}}x^{{\left(3\times 3\right)}^{2}}+\sum_{E_{\left(3∼4\right)}}x^{{\left(3\times 4\right)}^{2}}\;\\ & = & \left(24st+1\right)x^{9}+6\left(s+t-1\right))x^{36}+2\left(28st-2s-2t+1\right)x^{81}\;\\ & + & 4\left(2st-s-t+1\right)x^{144}. \end{array}$$Taking the first derivative of Equation (Equation 41) at x=1, we obtain the second *K*-hyper Banhatti index
(42)$$HB_{2}\left({\mathrm{MOF}}_{\left(s,t\right)}\right)=5904st-684s-684t+693.$$Now, we compute the second *K*-hyper Banhatti entropy of ${\mathrm{MOF}}_{\left(s,t\right)}$ by using Table 3 and Equation (Equation 42) in Equation (Equation 13) in the following way:
$$\begin{array}{ccc} ENT_{HB_{2}}\left({\mathrm{MOF}}_{\left(s,t\right)}\right) & = & \log\left(HB_{2}\right)-\frac{1}{HB_{2}}\log\{{\prod_{E_{\left(1,3\right)}}\left(v_{{\dot{a}}_{1}}\times v_{{\dot{a}}_{2}}\right)}^{2{\left(v_{{\dot{a}}_{1}}\times v_{{\dot{a}}_{2}}\right)}^{2}}\\ & \times & {\prod_{E_{\left(2,3\right)}}\left(v_{{\dot{a}}_{1}}\times v_{{\dot{a}}_{2}}\right)}^{2{\left(v_{{\dot{a}}_{1}}\times v_{{\dot{a}}_{2}}\right)}^{2}}\\ & \times & {\prod_{E_{\left(3,3\right)}}\left(v_{{\dot{a}}_{1}}\times v_{{\dot{a}}_{2}}\right)}^{2{\left(v_{{\dot{a}}_{1}}\times v_{{\dot{a}}_{2}}\right)}^{2}}\times {\prod_{E_{\left(3,4\right)}}\left(v_{{\dot{a}}_{1}}\times v_{{\dot{a}}_{2}}\right)}^{2{\left(v_{{\dot{a}}_{1}}\times v_{{\dot{a}}_{2}}\right)}^{2}}. \end{array}$$After simplification, we obtain
(43)$$\begin{array}{ccc} & = & \log\left(5904st-684s-684t+693\right)-\frac{1}{5904st-684s-684t+693}\log\{\left(24st+1\right)3^{18}\;\\ & \times & 6\left(s+t-1\right)6^{72}\times 2\left(28st-2s-2t+1\right)9^{162}\times 4\left(2st-s-t+1\right)12^{288}. \end{array}$$
**The first redefined Zagreb entropy of ${\mathrm{MOF}}_{\left(s,t\right)}$**
Let ${\mathrm{MOF}}_{\left(s,t\right)}$ be a metal–organic framework. Then, using Equation (Equation 5) and Table 3, the first redefined Zagreb polynomial is
(44)$$\begin{array}{ccc} ReZG_{1}\left({\mathrm{MOF}}_{\left(s,t\right)},x\right) & = & \sum_{E_{\left(1∼3\right)}}x^{\frac{1+3}{1\times 3}}+\sum_{E_{\left(2∼3\right)}}x^{\frac{2+3}{2\times 3}}+\sum_{E_{\left(3∼3\right)}}x^{\frac{3+3}{3\times 3}}+\sum_{E_{\left(3∼4\right)}}x^{\frac{3+4}{3\times 4}}\;\\ & = & \left(24st+1\right)x^{\frac{4}{3}}+6\left(s+t-1\right)x^{\frac{5}{6}}+2\left(28st-2s-2t+1\right)x^{\frac{2}{3}}\;\\ & + & 4\left(2st-s-t+1\right)x^{\frac{7}{12}}. \end{array}$$Taking the first derivative of Equation (Equation 44) at x=1, we obtain the first redefined Zagreb index
(45)$$ReZG_{1}\left({\mathrm{MOF}}_{\left(s,t\right)}\right)=2\left(37st+2\right).$$Now, we compute the first redefined Zagreb entropy using Table 3 and Equation (Equation 45) in Equation (Equation 14) in the following way:
$$\begin{array}{ccc} ENT_{ReZG_{1}}\left({\mathrm{MOF}}_{\left(s,t\right)},x\right) & = & \log\left(ReZG_{1}\right)-\frac{1}{ReZG_{1}}\log\{{\prod_{E_{\left(1,3\right)}}\left[\frac{v_{{\dot{a}}_{1}}+v_{{\dot{a}}_{2}}}{v_{{\dot{a}}_{1}}v_{{\dot{a}}_{2}}}\right]}^{\left[\frac{v_{{\dot{a}}_{1}}+v_{{\dot{a}}_{2}}}{v_{{\dot{a}}_{1}}d_{v}}\right]}\\ & \times & {\prod_{E_{\left(2,3\right)}}\left[\frac{v_{{\dot{a}}_{1}}+v_{{\dot{a}}_{2}}}{v_{{\dot{a}}_{1}}v_{{\dot{a}}_{2}}}\right]}^{\left[\frac{v_{{\dot{a}}_{1}}+d_{v}}{v_{{\dot{a}}_{1}}v_{{\dot{a}}_{2}}}\right]}\times {\prod_{E_{\left(3,3\right)}}\left[\frac{v_{{\dot{a}}_{1}}+v_{{\dot{a}}_{2}}}{v_{{\dot{a}}_{1}}v_{{\dot{a}}_{2}}}\right]}^{\left[\frac{v_{{\dot{a}}_{1}}+v_{{\dot{a}}_{2}}}{v_{{\dot{a}}_{1}}v_{{\dot{a}}_{2}}}\right]}\\ & \times & {\prod_{E_{\left(3,4\right)}}\left[\frac{v_{{\dot{a}}_{1}}+v_{{\dot{a}}_{2}}}{v_{{\dot{a}}_{1}}v_{{\dot{a}}_{2}}}\right]}^{\left[\frac{v_{{\dot{a}}_{1}}+d_{v}}{v_{{\dot{a}}_{1}}v_{{\dot{a}}_{2}}}\right]}\}. \end{array}$$After simplification, we obtain
$$\begin{array}{ccc} & = & \log2(37st+2)-\frac{1}{2(37st+2)}\log\{\left(24st+1\right){\left(\frac{4}{3}\right)}^{\frac{4}{3}}\times 6\left(s+t-1\right){\left(\frac{5}{6}\right)}^{\frac{5}{6}}\\ & \times & 2\left(28st-2s-2t+1\right){\left(\frac{6}{9}\right)}^{\frac{6}{9}}\times 4\left(2st-s-t+1\right){\left(\frac{7}{12}\right)}^{\frac{7}{12}}\}. \end{array}$$
**The second redefined Zagreb entropy of ${\mathrm{MOF}}_{\left(s,t\right)}$**
Let ${\mathrm{MOF}}_{\left(s,t\right)}$ be a metal–organic framework. Then, using Equation (Equation 6) and Table 3, the second redefined Zagreb polynomial is
(46)$$\begin{array}{ccc} ReZG_{2}\left({\mathrm{MOF}}_{\left(s,t\right)},x\right) & = & \sum_{E_{\left(1∼3\right)}}x^{\frac{1\times 3}{1+3}}+\sum_{E_{\left(2∼3\right)}}x^{\frac{2\times 3}{2+3}}+\sum_{E_{\left(3∼3\right)}}x^{\frac{3\times 3}{3+3}}+\sum_{E_{\left(3∼4\right)}}x^{\frac{3\times 4}{3+4}}\;\\ & = & \left(24st+1\right)x^{\frac{3}{4}}+6\left(s+t-1\right)x^{\frac{6}{5}}+2\left(28st-2s-2t+1\right)x^{\frac{3}{2}}\;\\ & + & 4\left(2st-s-t+1\right)x^{\frac{12}{7}}. \end{array}$$Taking the first derivative of Equation (Equation 46) at x=1, we obtain the second redefined Zagreb index
(47)$$ReZG_{2}\left({\mathrm{MOF}}_{\left(s,t\right)}\right)=\frac{810}{7}st-\frac{198}{35}\left(s+t\right)+\frac{198}{35}.$$Now, we compute the second redefined Zagreb entropy by using Table 3 and Equation (Equation 47) in Equation (Equation 15) in the following way:
$$\begin{array}{ccc} ENT_{ReZG_{2}}\left({\mathrm{MOF}}_{\left(s,t\right)},x\right) & = & \log\left(ReZG_{2}\right)-\frac{1}{ReZG_{2}}\log\{{\prod_{E_{\left(1,3\right)}}\left[\frac{v_{{\dot{a}}_{1}}v_{{\dot{a}}_{2}}}{v_{{\dot{a}}_{1}}+v_{{\dot{a}}_{2}}}\right]}^{\left[\frac{v_{{\dot{a}}_{1}}v_{{\dot{a}}_{2}}}{v_{{\dot{a}}_{1}}+v_{{\dot{a}}_{2}}}\right]}\\ & \times & {\prod_{E_{\left(2,3\right)}}\left[\frac{v_{{\dot{a}}_{1}}v_{{\dot{a}}_{2}}}{v_{{\dot{a}}_{1}}+v_{{\dot{a}}_{2}}}\right]}^{\left[\frac{v_{{\dot{a}}_{1}}v_{{\dot{a}}_{2}}}{d_{u}+v_{{\dot{a}}_{2}}}\right]}\times {\prod_{E_{\left(3,3\right)}}\left[\frac{v_{{\dot{a}}_{1}}v_{{\dot{a}}_{2}}}{v_{{\dot{a}}_{1}}+v_{{\dot{a}}_{2}}}\right]}^{\left[\frac{v_{{\dot{a}}_{1}}v_{{\dot{a}}_{2}}}{v_{{\dot{a}}_{1}}+v_{{\dot{a}}_{2}}}\right]}\\ & \times & {\prod_{E_{\left(3,4\right)}}\left[\frac{v_{{\dot{a}}_{1}}v_{{\dot{a}}_{2}}}{v_{{\dot{a}}_{1}}+v_{{\dot{a}}_{2}}}\right]}^{\left[\frac{v_{{\dot{a}}_{1}}v_{{\dot{a}}_{2}}}{v_{{\dot{a}}_{1}}+v_{{\dot{a}}_{2}}}\right]}\}. \end{array}$$After simplification, we obtain
$$\begin{array}{ccc} & = & \log\left(\frac{810}{7}st-\frac{198}{35}\left(s+t\right)+\frac{198}{35}\right)-\frac{1}{\frac{810}{7}st-\frac{198}{35}\left(s+t\right)+\frac{198}{35}}\log\{\left(24st+1\right){\left(\frac{3}{4}\right)}^{\frac{3}{4}}\times 6\left(s+t-1\right){\left(\frac{6}{5}\right)}^{\frac{6}{5}}\\ & \times & 2\left(28st-2s-2t+1\right){\left(\frac{9}{6}\right)}^{\frac{9}{6}}\times 4\left(2st-s-t+1\right){\left(\frac{12}{7}\right)}^{\frac{12}{7}}\}. \end{array}$$
**The third redefined Zagreb entropy of ${\mathrm{MOF}}_{\left(s,t\right)}$**
Let ${\mathrm{MOF}}_{\left(s,t\right)}$ be a metal–organic framework. Then, using Equation (Equation 7) and Table 3, the third redefined Zagreb polynomial is
$$\begin{array}{ccc} ReZG_{3}\left({\mathrm{MOF}}_{\left(s,t\right)},x\right) & = & \sum_{E_{\left(1∼3\right)}}x^{\left(1\times 3)(1+3\right)}+\sum_{E_{\left(2∼3\right)}}x^{\left(2\times 3)(2+3\right)}+\sum_{E_{\left(3∼3\right)}}x^{\left(3\times 3)(3+3\right)}\\ & + & \sum_{E_{\left(3∼4\right)}}x^{\left(3\times 4)(3+4\right)}\\ & = & \left(24st+1\right)x^{12}+6\left(s+t-1\right)x^{30}+2\left(28st-2s-2t+1\right)x^{54}\\ & + & 4\left(2st-s-t+1\right)x^{84}. \end{array}$$
(48)$$\begin{array}{ccc} ReZG_{3}\left({\mathrm{MOF}}_{\left(s,t\right)},x\right) & = & \left(24st+1\right)x^{12}+6\left(s+t-1\right)x^{30}+2\left(28st-2s-2t+1\right)x^{54}\;\\ & + & 4\left(2st-s-t+1\right)x^{84}. \end{array}$$Taking the first derivative of Equation (Equation 48) at x=1, we obtain the third redefined Zagreb index
(49)$$ReZG_{2}\left({\mathrm{MOF}}_{\left(s,t\right)}\right)=3984st-372\left(s+t\right)+384.$$Now, we compute the third redefined Zagreb entropy by using Table 3 and Equation (Equation 49) in Equation (Equation 16) in the following way:
$$\begin{array}{ccc} ENT_{ReZG_{3}}\left({\mathrm{MOF}}_{\left(s,t\right)},x\right) & = & \log\left(ReZG_{3}\right)-\frac{1}{ReZG_{3}}\log\{{\prod_{E_{\left(1,3\right)}}\left[\left(d_{u}v_{{\dot{a}}_{2}}\right)\left(d_{u}+v_{{\dot{a}}_{2}}\right)\right]}^{\left[\left(v_{{\dot{a}}_{1}}v_{{\dot{a}}_{2}}\right)\left(v_{{\dot{a}}_{1}}+v_{{\dot{a}}_{2}}\right)\right]}\\ & \times & {\prod_{E_{\left(2,3\right)}}\left[\left(v_{{\dot{a}}_{1}}v_{{\dot{a}}_{2}}\right)\left(v_{{\dot{a}}_{1}}+v_{{\dot{a}}_{2}}\right)\right]}^{\left[\left(d_{u}v_{{\dot{a}}_{2}}\right)\left(v_{{\dot{a}}_{1}}+v_{{\dot{a}}_{2}}\right)\right]}\\ & \times & {\prod_{E_{\left(3,3\right)}}\left[\left(v_{{\dot{a}}_{1}}v_{{\dot{a}}_{2}}\right)\left(v_{{\dot{a}}_{1}}+v_{{\dot{a}}_{2}}\right)\right]}^{\left[\left(v_{{\dot{a}}_{1}}v_{{\dot{a}}_{2}}\right)\left(v_{{\dot{a}}_{1}}+v_{{\dot{a}}_{2}}\right)\right]}\\ & \times & {\prod_{E_{\left(3,4\right)}}\left[\left(v_{{\dot{a}}_{1}}v_{{\dot{a}}_{2}}\right)\left(v_{{\dot{a}}_{1}}+v_{{\dot{a}}_{2}}\right)\right]}^{\left[\left(v_{{\dot{a}}_{1}}v_{{\dot{a}}_{2}}\right)\left(v_{{\dot{a}}_{1}}+v_{{\dot{a}}_{2}}\right)\right]}\}. \end{array}$$After simplification, we obtain
$$\begin{array}{ccc} \; & = & \log\left(3984st-372(s+t)+384\right)-\frac{1}{3984st-372(s+t)+384}\log\{\left(24st+1\right)12^{12}\\ \; & \times & 6\left(s+t-1\right)30^{30}\times 2\left(28st-2s-2t+1\right)54^{54}\times 4\left(2st-s-t+1\right)84^{84}\}. \end{array}$$
**Atom-bond sum connectivity entropy of ${\mathrm{MOF}}_{\left(s,t\right)}$**
Let NbO be a network of a niobium II oxide molecule. Then, using Equation (Equation 8) and Table 1, the atom-bond sum connectivity polynomial is
(50)$$\begin{array}{ccc} ABS({\mathrm{MOF}}_{\left(s,t\right)},x) & = & \sum_{E_{\left(1∼3\right)}}x^{\sqrt{\frac{1+3-2}{1+3}}}+\sum_{E_{\left(2∼3\right)}}x^{\sqrt{\frac{2+3-2}{2+3}}}+\sum_{E_{\left(3∼3\right)}}x^{\sqrt{\frac{3+3-2}{3+3}}}+\sum_{E_{\left(3∼4\right)}}x^{\sqrt{\frac{3+4-2}{3+4}}}\;\\ & = & \left(24st+1\right)x^{\frac{1}{\sqrt{2}}}+6\left(s+t-1\right)x^{\sqrt{\frac{3}{5}}}+2\left(28st-2s-2t+1\right)x^{\sqrt{\frac{2}{3}}}\;\\ & + & 4\left(2st-s-t+1\right)x^{\sqrt{\frac{5}{7}}}. \end{array}$$Taking the first derivative of Equation (Equation 50) at x=1, we obtain the atom-bond sum connectivity index
(51)$$\begin{array}{c} ABS\left(\mathrm{MOF}\right)=\left(24st+1\right)\frac{1}{\sqrt{2}}+6\left(s+t-1\right)\sqrt{\frac{3}{5}}+2\left(28st-2s-2t+1\right)\sqrt{\frac{2}{3}}\\ +4\left(2st-s-t+1\right)\sqrt{\frac{5}{7}}. \end{array}$$Now, we compute the third redefined Zagreb entropy using Table 3 and Equation (Equation 51) in Equation (Equation 17) in the following way:
$$\begin{array}{ccc} ENT_{ABS}\left(MOF\right) & = & \log\left(ABS\right)-\frac{1}{ABS}\log\{{\prod_{E_{\left(1,3\right)}}\left[\sqrt{\frac{\left(v_{{\dot{a}}_{1}}+v_{{\dot{a}}_{2}}-2\right)}{\left(v_{{\dot{a}}_{1}}+v_{{\dot{a}}_{2}}\right)}}\right]}^{\left[\sqrt{\frac{\left(v_{{\dot{a}}_{1}}+v_{{\dot{a}}_{2}}-2\right)}{\left(v_{{\dot{a}}_{1}}+v_{{\dot{a}}_{2}}\right)}}\right]}\;\\ \; & \times & {\prod_{E_{\left(2,3\right)}}\left[\sqrt{\frac{\left(v_{{\dot{a}}_{1}}+v_{{\dot{a}}_{2}}-2\right)}{\left(v_{{\dot{a}}_{1}}+v_{{\dot{a}}_{2}}\right)}}\right]}^{\left[\sqrt{\frac{\left(v_{{\dot{a}}_{1}}+v_{{\dot{a}}_{2}}-2\right)}{\left(v_{{\dot{a}}_{1}}+v_{{\dot{a}}_{2}}\right)}}\right]}\times {\prod_{E_{\left(3,3\right)}}\left[\sqrt{\frac{\left(v_{{\dot{a}}_{1}}+v_{{\dot{a}}_{2}}-2\right)}{\left(v_{{\dot{a}}_{1}}+v_{{\dot{a}}_{2}}\right)}}\right]}^{\left[\sqrt{\frac{\left(v_{{\dot{a}}_{1}}+v_{{\dot{a}}_{2}}-2\right)}{\left(v_{{\dot{a}}_{1}}+v_{{\dot{a}}_{2}}\right)}}\right]}\;\\ \; & \times & {\prod_{E_{\left(3,4\right)}}\left[\sqrt{\frac{\left(v_{{\dot{a}}_{1}}+v_{{\dot{a}}_{2}}-2\right)}{\left(v_{{\dot{a}}_{1}}+v_{{\dot{a}}_{2}}\right)}}\right]}^{\left[\sqrt{\frac{\left(v_{{\dot{a}}_{1}}+v_{{\dot{a}}_{2}}-2\right)}{\left(v_{{\dot{a}}_{1}}+v_{{\dot{a}}_{2}}\right)}}\right]}\}\;\\ \; & = & \log\left(ABS\right)-\frac{1}{ABS}\log\{\left(24st+1\right){\left(\frac{1}{\sqrt{2}}\right)}^{\frac{1}{\sqrt{2}}}\times 6\left(s+t-1\right){\left(\sqrt{\frac{3}{5}}\right)}^{\sqrt{\frac{3}{5}}}\;\\ \; & \times & 2\left(28st-2s-2t+1\right){\left(\sqrt{\frac{2}{3}}\right)}^{\sqrt{\frac{2}{3}}}\times 4\left(2st-s-t+1\right){\left(\sqrt{\frac{5}{7}}\right)}^{\sqrt{\frac{5}{7}}}\}. \end{array}$$

### Comparison

In this section, we compare the *K*-Banhatti and redefined Zagreb indices for ${\mathrm{MOF}}_{\left(s,t\right)}$ numerically and graphically in Table 4 and Figure 4, respectively.

## 5. Conclusions

The remarkable optical properties of metallic nanoparticles have piqued the interest of scientists and researchers. In this article, two important molecules niobium dioxide NbO${}_{2}$ and the ${\mathrm{MOF}}_{\left(s,t\right)}$ were considered, and the accurate formulas of some important valency-based topological indices were calculated using the technique of atom-bond partitioning. We investigated the distance-based entropies associated with a new information function and evaluated the relationship between degree-based topological indices and degree-based entropies in this article using Shannon’s entropy and Chen et al.’s entropy definitions. The idea of distance-based entropy is widely ingrained in industrial chemistry. It is used to calculate the complexity of molecules and molecular ensembles, their electronic structure, signal processing, physicochemical processes, and so on. The *K*-Banhatti entropy, in conjunction with the chemical structure, thermodynamic entropy, energy, and computer sciences can play an essential role in bridging various domains and providing a foundation for new interdisciplinary research. In the future, we hope to expand this concept to include various chemical structures, allowing researchers to pursue new avenues in this field. 

## Figures and Tables

**Figure 1 molecules-27-06975-f001:**
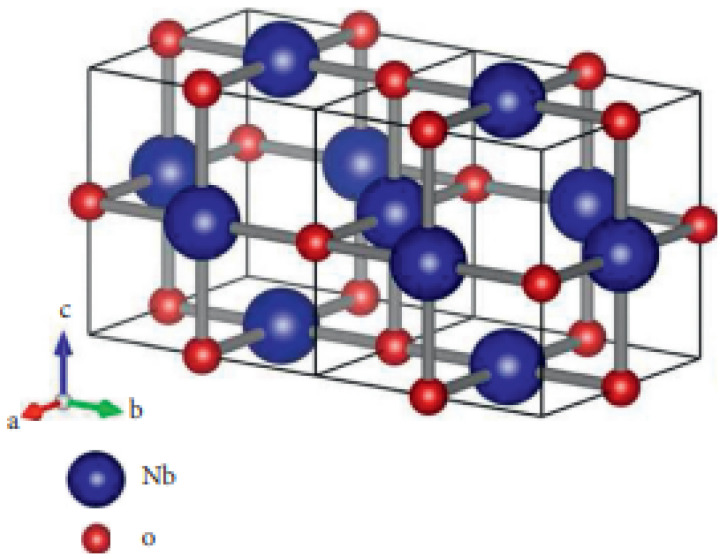
Niobium dioxide 3D structure.

**Figure 2 molecules-27-06975-f002:**
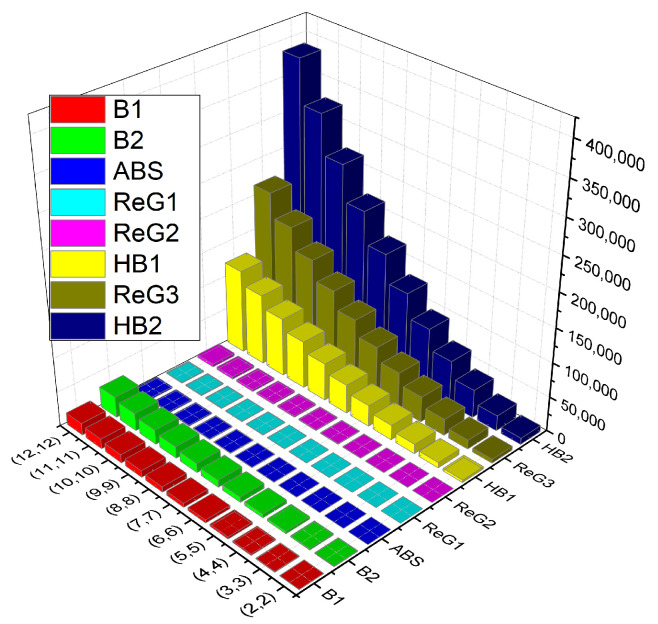
Graphical comparison of TI’s of NbO${}_{2}$.

**Figure 3 molecules-27-06975-f003:**
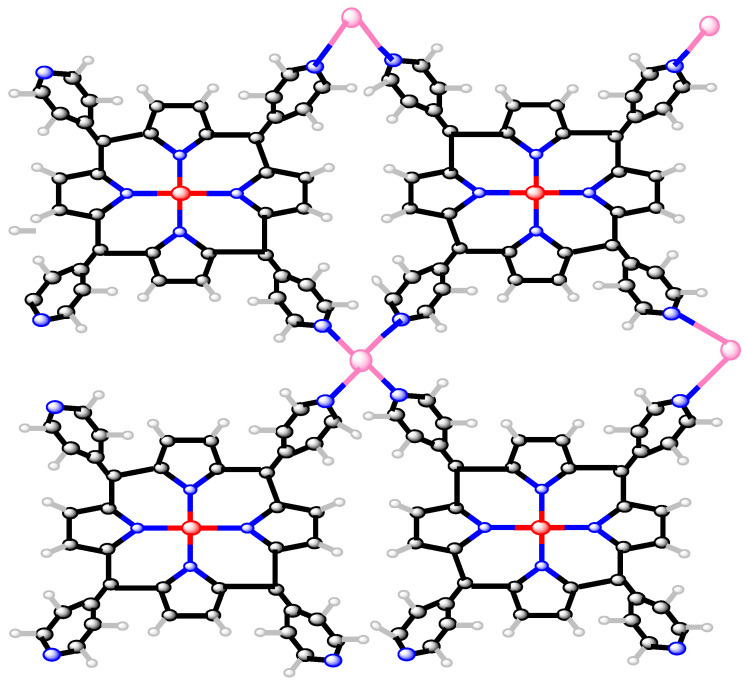
Two-dimensional MOF_(2,2) structure.

**Figure 4 molecules-27-06975-f004:**
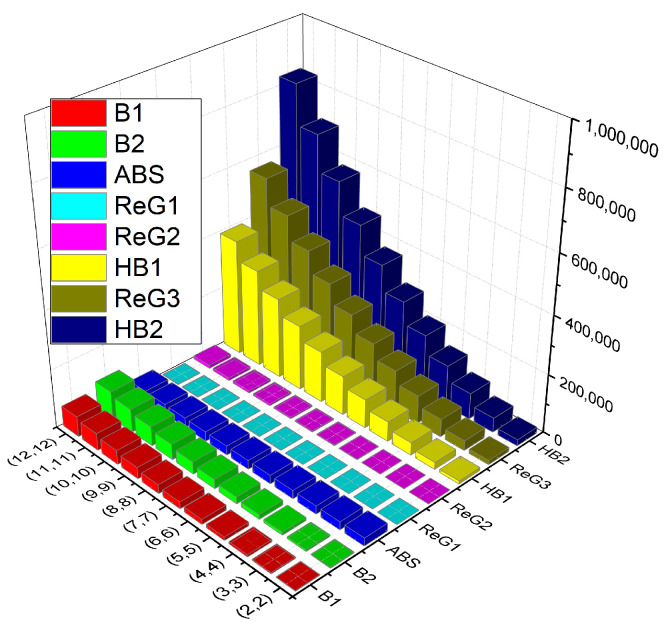
Graphical comparison of TI’s of metal–organic framework.

**Table 1 molecules-27-06975-t001:** Atom-bond partition of NbO${}_{2}$.

Types of Atom Bonds	$E_{\left(2∼3\right)}$	$E_{\left(3∼3\right)}$	$E_{\left(3∼4\right)}$	$E_{\left(4∼4\right)}$
Cardinality of Atom bonds	16	8(2s+2t−3)	4(3st−2s−2t+2)	2(2st-s-t)

**Table 2 molecules-27-06975-t002:** Numerical comparison of the *K*-Banhatti topological indices of NbO${}_{2}$.

(s,t)	$B_{1}$	$B_{2}$	${\mathrm{HB}}_{1}$	${\mathrm{HB}}_{2}$	${\mathrm{ReG}}_{1}$	${\mathrm{ReG}}_{2}$	${\mathrm{ReG}}_{3}$	*ABS*
(2,2)	552	872	3528	9320	58	136.34	5680	75.920117
(3,3)	1180	1944	7860	22,344	113	291.77	13,152	160.400806
(4,4)	2040	3432	13,880	408,872	186	504.34	23,664	275.748201
(5,5)	3132	5336	21,588	64,904	277	774.058	37,216	421.962304
(6,6)	4456	7656	30,984	94,440	386	1100.91	53,808	599.043115
(7,7)	6012	10,392	42,068	129,480	513	1484.9	73,440	806.990632
(8,8)	7800	13,544	54,840	170,024	658	1926.1	96,112	1045.804857
(9,9)	9820	17,112	69,300	216,072	821	2424.3	121,824	1315.48579
(10,10)	12,072	21,096	85,448	267,624	1002	2979.7	150,576	1616.03343
(11,11)	14,556	25,496	103,284	324,680	1201	3592.3	182,368	1947.447777
(12,12)	17,272	30,312	122,808	387,240	1418	4262.1	217,200	2309.728831

**Table 3 molecules-27-06975-t003:** Atom-bonds partition of ${\mathrm{MOF}}_{\left(s,t\right)}$.

Types of Atom Bonds	$E_{\left(1∼3\right)}$	$E_{\left(2∼3\right)}$	$E_{\left(3∼3\right)}$	$E_{\left(3∼4\right)}$
Cardinality of Atom bonds	(1+24st)	6(s+t−1)	2(28st−2s−2t+1)	4(2*st* − *s* − *t* + 1)

**Table 4 molecules-27-06975-t004:** Numerical comparison of the topological indices of ${\mathrm{MOF}}_{\left(s,t\right)}$.

(s,t)	$B_{1}$	$B_{2}$	${\mathrm{HB}}_{1}$	${\mathrm{HB}}_{2}$	${\mathrm{ReG}}_{1}$	${\mathrm{ReG}}_{2}$	${\mathrm{ReG}}_{3}$	ABS
(2,2)	1868	2529	10,542	21,573	296	307.03	14832	27,339.22
(3,3)	4264	5793	24,122	49,725	444	700.71	34008	27,686.67
(4,4)	7636	10,401	43,286	89,685	592	1256.40	61152	28,173.03
(5,5)	11,984	16,353	68,034	141,453	740	1974.09	96,264	28,798.29
(6,6)	17,308	23,649	98,366	205,029	888	2853.77	139,344	29,562.45
(7,7)	23,608	32,289	134,282	280,413	1036	3895.46	190,392	30,465.50
(8,8)	30,884	42,273	175,782	367,605	1184	5099.14	249,408	31,507.46
(9,9)	39,136	53,601	222,866	466,605	1332	6464.83	316,392	32,688.32
(10,10)	48,364	66,273	275,534	577,413	1480	7992.51	391,344	34,008.08
(11,11)	58,568	80,289	333,786	700,029	1628	9682.20	474,264	35,466.73
(12,12)	69,748	95,649	397,622	834,453	1776	11,533.89	565,152	37,064.29

## Data Availability

All data generated or analyzed during this study are included in this published article.

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
