# Peer review of "A Paradigmatic Approach to Find the Valency-Based K-Banhatti and Redefined Zagreb Entropy for Niobium Oxide and a Metal–Organic Framework"

_molecules, 2022, doi:10.3390/molecules27206975_

Round 1

Reviewer 1 Report

Various definitions of entropies are given. Topological indices are
computed for NbO and MOFs, and subsequently the entropy based on these indices is obtained.

I think there would be still quite a way to connect entropies as computed
here with the electronic structure or physical properties;
and I find these entropies not too interesting.

Some points to be addressed:

1) Equation 1, what is summed over (i.e. the meaning of the tilde-symbol
in the sum should be defined, is it some bonds/neighbors?)

2) Equation 8: I think on the right hand side, an overall minus is missing,
otherwise the entropy would be negative; and equation 9 only matches
when having this minus sign.

3) NbO: the parameters s and t need to be defined. Where does the
structure in figure 1 come from? Bulk NbO has a cubic structure with
space group 221, how does this fit in ?

4) Table 2: the acronyms B_1, B_2 ... should be explained. I assume
it refers to 1st K-Banhatti , 2nd K-Banhatti ... entropy, but this should
be defined.

5) MOF: again, how is s and t defined?

6) Table 3: how is 1,2,3,4 defined (e.g. 1 ~ 3 is which bond?)

7) references: better cite original papers - e.g. instead of [8] better
Wiener's paper form 1947

Author Response

Respected Sir

First of all, many thanks for your attention, kindness and for your careful review of our paper. The comments, corrections and suggestions from you are very valuable and helpful in revision and improvement of our manuscript. These comments provided us significant guidance to our research. We considered them carefully and have made following responses and revisions which we hope to meet the criteria. We wish that you are satisfied with the revision.

Reviewer 2 Report

The authors in this paper calculate several graph invariants (concerning the K-Banhatti indices, the 1st redefined Zagreb index, the 2nd redefined Zagreb index, and the third redefined Zagreb index) of the molecular graphs of Niobium Oxide and metal-organic. The authors may include similar results to the paper using Tables 1 and 3 for the atom-bond sum-connectivity index, introduced recently in https://doi.org/10.1007/s10910-022-01403-1

The redefined second Zagreb index is actually the same as the inverse sum indeg index that was defined in [D. Vukicevic, M. Gašperov, Croat. Chem. Acta 2010, 83, 243]. Also, the redefined first Zagreb index of every graph of order n is n. These observations are suggested to be included in the paper.

Author Response

Respected Sir,

First of all, many thanks for your attention, kindness and for your careful review of our paper. The comments, corrections and suggestions from you are very valuable and helpful in revision and improvement of our manuscript. These comments provided us significant guidance to our research. We considered them carefully and have made following responses and revisions which we hope to meet the criteria. We wish that you are satisfied with the revision.

Round 2

Reviewer 1 Report

The manuscript has been improved a bit, but still further improvement is necessary:

1) NbO: The authors write that it
"has a total atom count of 2 + 5s + 5t + 9st"
For s=t=1 , this would be 2+5+5+9 = 21 .
Why does one unit cell have an odd number of atoms, if the formula is NbO?

Further below, the authors write ": 2 atoms with valency 2,
8s + 8t + 4st − 8 atoms with valency 3, and 2 − 3s − 3t + 5st atoms with valency 4"

For s=t=1, this would be 2 atoms with valency 2, 12 atoms with valency 3,
and 1 atom with valence 4, in total 15 atoms, not 21 .

For very large s and t, considering only the term proportional st which
would be dominant, we would have 9st atoms according to one formula,
and 5st atoms according to the other formula.

This should be fixed/explained.

2) Please give the chemical formula of the MOF used.

Author Response

Response to the comments from respected reviewer

Manuscript Title: A Paradigmatic Approach to find the Valency Based K-Banhatti & Redefined Zagreb Entropy for Niobium Oxide and Metal-Organic Framework

First of all, many thanks for your attention, kindness and for your careful review of our paper. The comments, corrections and suggestions from you are very valuable and helpful in revision and improvement of our manuscript. These comments provided us significant guidance to our research. We considered them carefully and have made following responses and revisions which we hope to meet the criteria. We wish that you are satisfied with the revision.

Response to Reviewer #1 Second Round

  1. NbO: The authors write that it
    "has a total atom count of 2 + 5s + 5t + 9st"
    For s=t=1 , this would be 2+5+5+9 = 21 .
    Why does one unit cell have an odd number of atoms, if the formula is NbO?

    Further below, the authors write ": 2 atoms with valency 2,
    8s + 8t + 4st − 8 atoms with valency 3, and 2 − 3s − 3t + 5st atoms with valency 4"

    For s = t =1, this would be 2 atoms with valency 2, 12 atoms with valency 3,
    and 1 atom with valence 4, in total 15 atoms, not 21 .

    For very large s and t, considering only the term proportional st which
    would be dominant, we would have 9st atoms according to one formula,
    and 5st atoms according to the other formula.

    This should be fixed/explained.

Author Reply:

Dear Sir we are very thankful to you for your keen observation. We have addressed all the comments carefully while revising the manuscript. The responses to the comments point-wise are listed as follows:

We draw the structure again and analysis carefully due to the typo mistake initially we wrote NbO in some places. We find that the total number of atoms have 21 in niobium dioxide NbO2 when s = t = 1.

There are three types of atoms in NbO2 based on their valency: 8 atoms with valency 2, 8s + 8t + 4st − 8 atoms with valency 3, and 2 − 3s − 3t + 5st atoms with valency 4. Table 1 shows the atom-bond partitions of NbO2 derived from these results. Now we correct this in the whole manuscript.

NbO2 has a total atom count of 2 + 5s + 5t + 9st while 2 − 3s − 3t + 5st atoms with valency 4. In this paper we are taking the case when s = t as shown in Table 2.

  1. Please give the chemical formula of the MOF used.

Author Reply:

Dear Sir we are very thankful to you for your keen observation.

The metal-organic framework has the molecular formula FeTPyP − Co, where Fe denotes iron, TPyP denotes tetrakis pyridyl porphyrin, and Co denotes cobalt.

Moreover, after adding numerical values in Table 2 and Table 3 for atom-bond sum connectivity index for NbO2 and MoF, we redraw Figure 2 and Figure 4.
